# Silicon Nitride Ceramics: Structure, Synthesis, Properties, and Biomedical Applications

**DOI:** 10.3390/ma16145142

**Published:** 2023-07-21

**Authors:** Robert B. Heimann

**Affiliations:** Am Stadtpark 2A, D-02826 Görlitz, Germany; robert.heimann@ocean-gate.de

**Keywords:** silicon nitride, structure, processing, properties, biomedical application

## Abstract

Silicon nitride ceramics excel by superior mechanical, thermal, and chemical properties that render the material suitable for applications in several technologically challenging fields. In addition to high temperature, high stress applications have been implemented in aerospace gas turbines and internal combustion engines as well as in tools for metal manufacturing, forming, and machining. During the past few decades, extensive research has been performed to make silicon nitride suitable for use in a variety of biomedical applications. This contribution discusses the structure–property–application relations of silicon nitride. A comparison with traditional oxide-based ceramics confirms that the advantageous mechanical and biomedical properties of silicon nitride are based on a high proportion of covalent bonds. The present biomedical applications are reviewed here, which include intervertebral spacers, orthopedic and dental implants, antibacterial and antiviral applications, and photonic parts for medical diagnostics.

## 1. Introduction

At present, silicon nitride ceramics are being used predominately for system parts subjected to high temperatures and high stresses [1]. In aerospace and automotive applications, silicon nitride is used for wear- and corrosion-resistant engine parts and accessory units, including turbochargers, glow plugs for diesel engines, exhaust gas control valves, and rocker arm pads for gas engines. Other popular applications include a variety of molds, dies, and cutting tools for metal manufacturing, forming, and machining, as well as roller and ball bearings for machine tools and vacuum pumps [2,3,4]. These applications are based on the exceptional properties of silicon nitride, which include its low density, high elastic modulus and bending strength, reasonable fracture toughness up to 1400 °C, high wear and solid particle erosion resistance, low coefficient of thermal expansion, high thermoshock resistance, and stability against most acids and bases, corrosive gases, and liquid metals.

In addition, thin amorphous silicon nitride films deposited by chemical vapor deposition (CVD) and related technologies are used as effective masking layers and diffusion barriers during the production of microelectronic semiconductor chips [5], gate dielectrics for specific metal–insulator–semiconductor memory devices, moisture barriers for organic light-emitting diode displays, and photonic integrated circuits [6]. Futuristic geoengineering installations include positioning myriads of thin silicon nitride tiles in outer space at Lagrange point 1. These tiles are envisaged to act as effective solar reflectors, combating global warming by providing sunshade [7].

Apart from these uses, recent research has amply confirmed that the superior mechanical and biological properties of silicon nitride bulk structures and coatings can be utilized advantageously for a variety of biomedical devices and applications [8]. These include porous intervertebral spacers for spinal fusion surgery [9], mechanically resilient thin-walled hip resurfacing prostheses [10], high-strength dental implants [11,12], long-lasting micro-bearings for dental drills, and others. Since silicon nitride surfaces elicit antimicrobial properties [13], they have the potential to become major players in future hospital-based hygienic management to destroy multidrug-resistant organisms, including *Staphylococcus aureus*. Most recently, silicon nitride has been identified as powerful killer of the SARS-CoV-2 virus and other zoonotic organisms [14].

## 2. Crystallographic Structure

Under ambient conditions, silicon nitride crystallizes in two polymorphs, α- and β-Si_3_N_4_. The α-Si_3_N_4_ occurs naturally as the rare mineral nierite in some types of meteorites [15]. A third γ-modification with spinel structure is stable only at high pressure and temperatures [16]. The α-Si_3_N_4_ phase stable at low temperatures converts to β-Si_3_N_4_ at temperatures beyond 1.500 °C.

The space group of trigonal α-Si_3_N_4_ is *P31c* (159) with lattice dimensions *a* = 775.193(3) pm and *c* = 561.949(4) pm [17]. The space group of hexagonal β-Si_3_N_4_ is *P6_3_/m* (176) with lattice dimensions *a* = 760.8 pm and *c* = 291.1 pm. As shown in Figure 1, the structure consists of slightly distorted corner-sharing SiN_4_ tetrahedra that form hexagonal rings arranged in layers with ABCDABCD… stacking (α-Si_3_N_4_) and ABAB… stacking (β-Si_3_N_4_) modes. In the α-Si_3_N_4_ structure, the bond lengths vary between 156.6 pm (N1-Si1) and 189.6 pm (N1-Si2). In β-Si_3_N_4_, the bond lengths between Si and N1 are 173.0 and 173.9 pm, respectively. The bond length between Si and N2 is 174.5 pm. 

In both phases, the structures of the AB layers are identical. However, the CD layer in the α-phase is related to AB by a c-glide plane. Hence, the double layer in α-Si_3_N_4_ can be considered a superposition of a β-Si_3_N_4_ layer and its counterpart inverted by 180°. Consequently, as shown in Figure 1 there are twice as many atoms per unit cell in α-Si_3_N_4_ (Z = 4, right) than in β-Si_3_N_4_ (Z = 2, left). The Si_3_N_4_ tetrahedra in β-Si_3_N_4_ are interconnected in such a way that wide channels are formed that extend parallel to the c-axis of the unit cell. Owing to the c-glide plane, the α-structure contains two (isolated) interstitial sites per unit cell instead of channels. The channel structure of the β-phase supports the easy diffusion of ions through the lattice thus promoting enhanced sinterability at high processing temperatures. Hence, the sintering of load-carrying devices with their required high density should advantageously start from β-Si_3_N_4_ precursor powders.

## 3. Synthesis, Processing, and Properties of Silicon Nitride

### 3.1. Silicon Nitride Powders

The synthesis of silicon nitride powders can be achieved in several ways that involve various silicon carriers, such as elemental silicon, silicon dioxide, gaseous silicon tetrachloride and monosilane, as well as organo-silicon precursors [19]. The routes toward silicon nitride powders include the following: -Direct nitriding of silicon powder (see Figure 2);-Carbothermal nitriding of silicon dioxide;-Diimide synthesis by reacting silicon tetrachloride with ammonia;-Chemical vapor deposition synthesis using silicon tetrachloride or monosilane;-Plasma chemical synthesis, such as plasma-enhanced chemical vapor deposition;-Excimer laser-induced reactions of silane–ammonia mixtures;-Pyrolysis of organo-silicon compounds;-Sol-gel processing of polymeric precursors;-Self-propagating high-temperature synthesis.

**Figure 2 materials-16-05142-f002:**
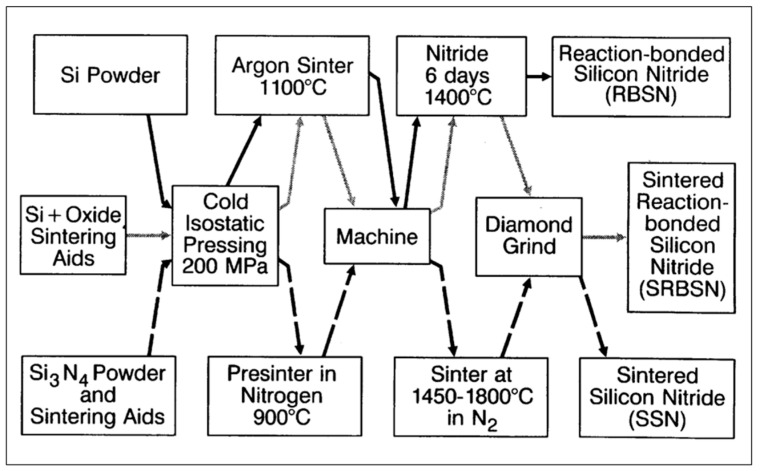
Flow chart of silicon nitride processing routes to form RBSN, SRBSN, and SSN compacts [18]. Solid arrows refer to RBSN, light arrows to SRBSN, and dashed arrows to SSN. © With permission from Wily-VCH, Weinheim, Germany.

The manufacturing and application of nano-sized silicon nitride powders are emerging as one of the fastest growing segments of the contemporary ceramic industry. Nanoparticles are defined as particles that have at least one dimension less than 100 nm. Given their small size, nanoscale particles have a high proportion of atoms close to their surfaces. Hence, strong deviations from the behavior of the bulk structure including chemical composition and reactivity are to be expected. The size also affects other important bulk properties based on the restriction on wavefunction radius, separation of lattice defects, and interacting strain fields [20]. To obtain commercial use, nanoscale silicon nitride requires the attainment of a complex set of properties that are listed in Table 1. 

Manufacturing routes of silicon nitride nanoparticles include attrition milling, sol-gel processing, vapor technologies, Joule–Thomson expansion, and other techniques [18].

### 3.2. Silicon Nitride Compacts

The silicon nitride powders synthesized by one of the routes mentioned above are further processed to form shaped compacts. Depending on the processing routes, there exist several structural and textural variants of silicon nitride that differ markedly in their mechanical, elastic, and thermal properties (Table 2). The variants most frequently used in engineering applications are porous reaction-bonded silicon nitride (RBSN) [21,22], dense sintered silicon nitride (SSN), as well as a combination of both, sintered reaction-bonded silicon nitride (SRBSN). These processing routes are schematically shown in Figure 2. Whereas the synthesis of RBSN and SRBSN starts from silicon powders that are sintered in argon prior to nitriding, the production of SSN avoids this reaction step. Furthermore, all three types of silicon nitride require machining of the final parts, but RBSN does not require diamond grinding, making parts fashioned from RBSN economically highly competitive.

#### 3.2.1. Reaction-Bonded Silicon Nitride (RBSN)

To obtain RBSN monolithic bodies, direct nitriding of fine compacted elemental silicon powder is employed that leads to the formation of a mixture of α- and β-Si_3_N_4_ by prolonged heating in nitrogen or ammonia atmospheres at temperatures up to 1400 °C. The fine silicon powder is nitrided predominately by a vapor phase reaction [2] that involves the oxidation of gaseous Si(g) to form gaseous silicon monoxide SiO(g) according to Equation (1). This SiO further reacts with nitrogen to form silicon nitride according to Equation (2).
2Si(g) + O_2_(g) → 2SiO(g)(1)
3SiO(g) + 2N_2_(g) →Si_3_N_4_(c) + 3/2O_2_(g)(2)

Since a volume increase of about 22% is completely accommodated by the inter-particle void space of the compacted silicon powder, no shrinking occurs during nitriding. Hence, the original dimensions of the green ceramics will be faithfully retained. This is the reason for using RBSN to shape components with complex geometry. Since the processing route does not require expensive diamond grinding, parts manufactured from RBSN are economically highly competitive. Although the porosity of RBSN ranges from 20 to 30%, its mechanical performance is remarkable as flexural strengths in the range of 200 to 400 MPa can be achieved. This strength level can be retained to about 1400 °C, so that designing with RBSN leads to monolithic ceramics with high Weibull moduli and associated high reliability in service.

#### 3.2.2. Sintered Silicon Nitride (SSN)

To obtain fully dense sintered silicon nitride (SSN) monolithic bodies, hot pressing or hot isostatic pressing of silicon nitride powder with added metal oxides is employed. The oxides most often used as sintering aids are magnesium or yttrium oxides. At temperatures above 1550 °C, these oxidic additives form with contaminant silicon dioxide films around individual silicon nitride grains a liquid siliceous binder phase of SiAlON (Si_6-z_Al_z_O_2_N_8-z_) composition in which silicon nitride readily dissolves. This intergranular, essentially glassy binder phase leads to the efficient densification of the sintered ceramic body. However, some precautions must be taken to limit the thermal decomposition of the silicon nitride or the loss of the additives by partial evaporation. To avoid this, the compact is sintered in a bed of silicon nitride powder and/or under a high-pressure nitrogen atmosphere (typically 1 to 8 MPa). The protective atmosphere suppresses the evaporation of silicon, nitrogen, and additives. To obtain fully dense SSN, the use of nanoscale silicon nitride powder and high temperatures of 1700 to 1850 °C are advised.

#### 3.2.3. Sintered Reaction-Bonded Silicon Nitride (SRBSN)

To reduce the final porosity of RBSN and thus to further improve its mechanical and elastic properties, SRBSN has been developed. More recent developments of sintered reaction-bonded silicon nitride (SRBSN) have shown improvements in toughness and impact resistance through the growth of crack-arresting or -deflecting rod-like silicon nitride grains [23]. In addition, depending on the nature of the oxidic sintering aids and the processing condition, high thermal conductivities can be attained [24]. The highest values of thermal conductivity were found by adding AlN to Y_2_O_3_ as a sintering aid that together with impurity oxygen dissolved in the lattice of β-Si_3_N_4_ and formed a β-SiAlON solid solution, i.e., Si_6−_*_z_*Al*_z_*O*_z_*N_8−_*_z_*. Increasing the sintering time increased the fracture toughness [23], as shown in Table 2.

#### 3.2.4. Textured Sintered Silicon Nitride (TSSN)

In addition, highly mechanically resilient textured sintered silicon nitride (TSSN) monolithic shapes were developed. The α-Si_3_N_4_ phase stable at low temperature converts to β-Si_3_N_4_ at the high sintering temperatures applied during the densification of powder compacts. The rod-like β-Si_3_N_4_ particles yield by so-called texturing a self-reinforcing microstructure with greatly enhanced flexural strength and modulus of elasticity (Table 2). Seeds of rod-like β-Si_3_N_4_ crystals are added to the raw powder consisting predominately of α-Si_3_N_4_. These crystals are aligned by hot working techniques such as hot pressing, hot forging, and sinter forging, or template grain growth involving alignment by cold pressing, extrusion of a slurry, tape casting, and static or rotating magnetic fields. The aligned β-Si_3_N_4_ seeds grow during subsequent annealing at high temperature into a reinforcing microstructure [25]. Although there is a tendency that large, elongated grains act as structural flaws that reduce flexural strength, careful tailoring of size, amount, and orientation of grains can alleviate this drawback [26].

### 3.3. Silicon Nitride Foams

Porous silicon nitride [21] excels by a unique combination of various mechanical, thermal, chemical, and electric properties such as its light weight, high strain and damage tolerance, high hardness and wear resistance, low thermal expansion coefficient, high-temperature resistance, good thermal shock resistance, oxidation and corrosion resistance, good biocompatibility, and favorable dielectric properties. Its fabrication routes include pressureless sintering [27], freeze drying [28], gel casting [29,30], the addition of sacrificial pore formers [31], the use of replica templates [32], three-dimensional printing [33], slip [34] and tape casting [35], microwave sintering [36], and other techniques.

For example, gel casting of Si-PMMA suspensions and subsequent nitriding and sintering produced SRBSN foams with a porosity up to 80 vol% [37]. The foams were found to be whisker-free and showed high pore interconnectivity after sintering. The foam struts consisted of β-Si_3_N_4_ grains that were embedded in an amorphous phase. Biocompatible silicon nitride foams may have potential biomedical applications that include key components for tissue substitution for in vitro and in vivo tissue engineering applications, including scaffolds for bone reconstruction [29] and for biosensing and medical diagnostics [38]. Based on the remarkable strength of silicon nitride, the application of silicon nitride foams loaded with polymers such as poly(methylmethacrylate, PMMA), poly(etheretherketone, PEEK), or poly(lactic acid, PLA) as intervertebral disk implants may be envisaged.

### 3.4. Silicon Nitride Coatings

Surface coatings have emerged as important tools of increasingly sophisticated surface engineering technologies. During the past decades, a reliance on functionality and economic considerations have propelled coatings into the limelight of materials engineering. Cheap but mechanically and thermally strong substrates are being functionally improved, protected, and refined by adding expensive but low-volume materials with exceptional properties that render the tandem substrate coating a synergistic success.

Consequently, many attempts have been made in the past to design silicon nitride coatings. However, the deposition of pure, i.e., unalloyed silicon nitride coatings by conventional thermal spraying techniques has been found to be impossible since Si_3_N_4_ does not melt congruently but decomposes and in turn sublimates above 1850 °C. It also oxidizes readily at elevated temperatures. 

On the one hand, thin amorphous silicon nitride films are successfully deposited by chemical vapor deposition (CVD) and applied as masking layers for semiconductor integrated circuits during profile etching, as diffusion barriers in very large-scale integrated production lines, and for the damage protection of optical fibers. Thin silicon nitride films were also deposited by radio-frequency magnetron sputtering [39] and other synthesis methods.

On the other hand, attempts to deposit mechanically stable thick silicon nitride coatings for non-electronic applications by thermal spraying using metallic [40] or silicate glass binders [41], conversion by a reactive spray process [42], or in situ nitridation in flight [43] were not very promising. It turned out that such coatings contain only a small percentage of unaltered silicon nitride but instead substantial amounts of embrittling metal silicides. More successful were attempts to deposit coatings with enhanced Si_3_N_4_ content starting from SiAlON precursor powders [44] or clad-type powder consolidation using alloy bond coats [45]. 

However, later it was recognized that high particle velocities generated by detonation spraying, Top Gun™ technology, high-frequency pulse detonation, or atmospheric plasma spraying with axial powder injection were essential to deposit dense and well-adhering coatings with high silicon nitride contents [46,47]. In addition, the optimization of heat transfer into powder particles to control the critical viscosity of the oxide binder phase was found to be one of the most decisive factors [48] that requires sophisticated powder preparation procedures [49].

Extremely high powder particle velocities up to 3 km/s, generated by an electromagnetically accelerated plasma (EMAP) [50], were applied to deposit dense, homogeneous silicon nitride coatings with desirable mechanical properties that adhere very tightly (adhesion strength > 75 MPa) to polished stainless steel surfaces [51]. The operating principle of EMAPs is shown in Figure 3. The deposition system comprises an evacuated accelerating channel that forms a pair of parallel electrodes connected to a high current power source by a switch. A separate vessel contains a pressurized gas and the powder mix. The process is initiated by activating a fast-opening valve at the nozzle of the vessel (Figure 3a). After introducing the powder and the working gas into the accelerating channel (Figure 3b), an arc discharge is initiated at the desired position in the accelerating channel. During discharge, the powder remains suspended in the accelerating channel (Figure 3c). The plasma at the arc initiating point then receives an electromagnetic force pulse and forms an electromagnetically accelerated plasma of the working gas that heats and propels the powder particles with supersonic speed towards the substrate (Figure 3d,e). 

Figure 4 shows dense and mechanically highly stable silicon nitride coatings on stainless steel substrates deposited by detonation (A) and EMAP spraying (B). Both types of coating are dense, although the porosity of the detonation-sprayed coating appears to be higher, presumably related to the comparatively large grain size range of +32–45 µm, in contrast to the powder with a grain size range of +8–20 µm used for EMAPs. It is remarkable that the EMAP-sprayed coating did not require substrate roughening to adhere very well. Indeed, the SUS304 steel substrate was mirror-polished prior to spraying to yield an adhesion strength exceeding 75 MPa (Figure 4B). In contrast, the detonation-sprayed coating needed roughening that resulted in the undulating substrate–coating interface shown in Figure 4A.

The powders used for detonation and EMAP spraying were synthesized by mixing commercial α-Si3N4 powder with oxidic sintering aids such as alumina and yttria, agglomerated by spray drying using an organic binder and subsequently sintered at 1450 °C in a nitrogen atmosphere [46,51]. The powders used for detonation (Figure 4A) and EMAP spraying (Figure 4B) contained 68 mass% Si3N4 + 16 mass% Al2O3 + 16 mass%, whereby the volume of the oxidic binder phase comprised 30%. The approximate composition of the resulting SiAlON binder phase was found to be Si_3_Al_3_O_3_N_5_ (z = 3) [46] and Si_5_AlON_7_ (z = 1) [51] for detonation- and EMAP-sprayed coatings, respectively.

The complex high-temperature reactions are shown in Equations (3) and (4). During spraying, silicon nitride reacts with the oxides of the binder phase at grain boundaries according to
Si_3_N_4_ + Al_2_O_3_ + Y_2_O_3_ → ß’-Si_6−z_Al_z_O_z_N_8−z_ + Y-Si-Al-O-N glass (I)(3)

During cooling, glass (I) devitrifies to form crystalline yttrium aluminum garnet (YAG) and glass (II). For example, for z = 1 this reaction can be expressed by
Si_5_AlON_7_ + glass(I) → Si_5+x_Al_1−x_O_1−x_N_7+x_ + Y_3_Al_5_O_12_ + Si-O-N glass (II).(4)

It ought to be mentioned that Equations (3) and (4) are mere simplified representations of the complex high-temperature reactions occurring in the ternary system Si_3_N_4_-Al_2_O_3_-Y_2_O_3_, and that the actual reaction process is thought to be dependent on several factors, including temperature, composition, grain size of the reacting constituents, and porosity and pore structure, including interconnectivity.

Figure 5 shows the phase composition of EMAP powder #1 (top) and the resulting coating (bottom). The coating consists prominently of α-Si_3_N_4_, minor amounts of ß-Si_3_N_4_, and traces of YAG and SiAlONs with different degrees of substitution 1 < z < 4, predominately z = 1. 

The high adhesion strength of the silicon nitride coatings to the mirror-polished stainless-steel substrate is related to the high-velocity impact of solid particles at the substrate surface that causes a mechanism akin to explosive cladding [52]. Solid or only partially molten particles impacting the substrate with supersonic velocity generate a shock wave that propagates into the substrate. Since the P-V adiabat, i.e., the Rankine–Hugoniot equation of the state of the porous particle, lies above that of dense material, its thermal energy increases by compression of the porous material. This is accompanied by an increase in the thermal pressure component. The propagating shock wave causes multiple collisions among crystal grains within the porous particle, and thus a high local pressure is generated that causes additional compression, isentropic heating, and crushing of the particle [53].

However, as such EMAP coatings appear successful from a materials science viewpoint, from a process economy perspective they are in dire need of improvement. The excellent coating performance notwithstanding, the batch-type mode of EMAPs with the required replacement of the powder and gas feeding vessel after each shot renders this technique uneconomical at present. Hence, process scale-up and optimization must be carried out during future development cycles with the aim to provide a continuous injection mode of the pressurized gas and the powder particles. 

## 4. Biomedical Applications

Silicon nitride is an emerging bioceramic material with unique bulk and surface properties that is destined to become a clinically relevant part of the biomedical engineering toolbox for applications in a variety of medical fields. These applications include intervertebral spacers, hip and knee endoprosthetic joint implants, bone grafts and scaffolds, dental implants, antibacterial and antiviral particles and coatings, and medical diagnostic tools and intelligent neural circuits [8,54,55]. All these applications are based on its high compressive and flexural strengths, high fracture toughness, low friction coefficient, high wear and corrosion resistance, low dielectric loss, improved medical imaging capability as a radiolucent material, enhanced biocompatibility and osseointegration, and antimicrobial activity. 

Because of its high proportion of covalent bonds, silicon nitride stands up well to the corrosive environment of the human body, in particular to the Cl-rich extracellular fluid that was found to be corrosive for many metallic biomaterials [56]. The resistance of silicon nitride against aqueous corrosion has been ascribed to the formation of a thin protective silicon oxide film as well as the presence of silica at grain boundaries [57]. 

Hence, based on its outstanding corrosion resistance and biocompatibility, silicon nitride takes on a hybrid role between bioinert and bioactive ceramics, bridging the conceptual gap that divides these functionally different materials [58].

### 4.1. Intervertebral Spacers

Polyetheretherketone (PEEK) is commonly used as an intervertebral spacer for spinal fusion surgery. However, PEEK is bioinert and does not effectively integrate into living bone. In contrast, spacers made of silicon nitride excel by a surface nanostructure and chemistry that encourage appositional bone healing [59]. Osseoconductive silicon nitride has been cleared by the US Food and Drug Administration for use in spinal intervertebral arthrodesis devices. Hence, silicon nitride has been used in porous intervertebral spacers for spinal fusion without any undesirable side effects for almost 40 years [60,61,62,63]. In particular, long-term assessment of the outcome of anterior lumbar interbody fusion has shown encouraging results as in all cases complete osseointegration without any reported inflammatory response was achieved [64]. Based on the remarkable strength of silicon nitride, the application of novel silicon nitride foams loaded with polymers such as PMMA, PEEK, or PLA as intervertebral disk implants may be envisaged.

### 4.2. Orthopedic Applications

Wear debris produced by articulating compounds of joint endoprostheses currently limits the long-term performance and safety of such devices. Cytotoxicity, inflammatory cytokine release, oxidative stress, and the genotoxicity potential of silicon nitride particles were evaluated using peripheral blood mononuclear cells from individual human donors [65]. The study demonstrated that silicon nitride is an attractive orthopedic biomaterial. Compared to both CoCr and Ti6Al4V, wear particles released from silicon nitride have minimal biological impact on human cells.

The strength of silicon nitride with a high degree of covalent bonding was found to be significantly higher than that of structural oxide ceramics with largely ionic bonds. A finite element analysis of silicon nitride hip resurfacing (HR) prostheses showed that the stress distribution within the femoral bone implanted with silicon nitride prostheses was comparable to an intact femoral bone. Lifetime predictions indicated that silicon nitride implants are mechanically reliable and thus suitable for HR prostheses [10]. Consequently, silicon nitride coatings are currently under investigation as bearing surfaces for joint implants, owing to their low wear rate and excellent biocompatibility [66], as well as substantially improved antibacterial and osteoinductive properties compared with Ti and PEEK [67].

The mechanical properties of silicon nitride (Table 2), associated with high biocompatibility and promising tribological features including a low friction coefficient, high fracture toughness, and high wear resistance, render it a promising candidate for replacing joint components [68]. Based on the superior resistance of silicon nitride to mechanical stress and wear [69], applications for compression-loaded implants such as knee and hip endoprostheses are suggested [70]. In addition to its biocompatibility, silicon nitride is known to have a surface chemistry favorable for osteogenesis and increased bone-to-implant contact, resulting in the improved structural and functional osseointegration between the living bone and the surface of an artificial implant [71]. In vitro tribological tests revealed an extremely low coefficient of friction that was found to decrease with increasing articulating time. This advantageous behavior has been attributed to hydrodynamic lubrication by a hydrogen-terminated silicon oxide boundary film that develops during frictional loading and separates the articulating components [72].

However, there is a catch: the high modulus of elasticity of both hot-pressed (HPSN) and pressureless sintered (SSN) silicon nitride exceeding 300 GPa is counter indicative of their use in hip implants. This is based on the risk of strong stress shielding that occurs when the loading stress during walking is taken up by the high modulus implant materials instead of the bone structure. Since bone must be loaded in tension to remain healthy, stress shielding may lead to bone resorption by osteolysis and impaired remodeling according to Wolff’s law. As an alternative, RBSN may be used, which has a considerably lower modulus around 150 GPa. However, this value is still much higher than that of cortical bone (around 20 GPa). Furthermore, both the flexural strength (150–350 MPa) and fracture toughness (1.5–3 MPa·√m) of RBSN are much lower than those of HPSN and SSN (500–1.000 MPa and 5–8 MPa·√m, respectively), a consequence of the high porosity of the former [18].

Despite this drawback, acetabular cups based on silicon nitride have already found their way into the clinical operation theater [73]. The reason for this relies on the fact that in most hip implant systems, the metallic or ceramic femoral head articulates against an acetabular cup lined with tough polymers such as ultra-high-molecular-weight polyethylene or X-linked polyethylene. Unfortunately, these polymers degrade with time, and thus strategies must be developed to delay or even avoid this deterioration. Based on the unique surface chemistry of silicon nitride, the material can absorb hydrogen released from degrading polyethylene, thus limiting acetabular liner degradation. This also reduces the risk of polymeric wear debris known to cause the so-called ‘particle disease’ that may lead to aseptic loosening of the implant by periprosthetic osteolysis.

In a recent study, non-stoichiometric silicon nitride (SiN_x_) coatings were deposited by reactive high-power impulse magnetron sputtering on CoCrMo alloy surfaces with an adhesion-promoting CrN interlayer bond coat. Wear tests performed against Si_3_N_4_ in a reciprocating ball-on-disc test revealed specific wear rates lower than or comparable to CoCrMo [74].

Silicon nitride coatings were deposited by cathodic arc deposition on Ti6Al7Nb surfaces and subsequently treated with a Yb-doped fiber laser beam [75]. The laser irradiation improved the microhardness and reduced the residual stress and the coefficient of friction. Hence, the improvement of the wear performance of silicon nitride coatings by laser treatment suggests successful application in the orthopedic realm.

### 4.3. Craniofacial Reconstruction

Damaged bone could be filled with dedicated plugs fashioned from silicon nitride foam that provides a template for newly formed osteocytes. Furthermore, because the surface properties of silicon nitride promote bone ongrowth and ingrowth, it has the potential to benefit craniofacial reconstruction. The Si_3_N_4_ surface enhances osteoprogenitor cell adhesion and growth, specific expression of runt-related transcription factor 2 (RUNX2), and formation of extracellular matrix through the coupled effects of higher surface energy and the presence of amide and nanocrystalline hydroxylapatite functional groups [76]. Surface nitrogen appears to induce the formation of N-H moieties acting as precursors to the amide groups present in collagenous extracellular matrix (ECM). These moieties elevate the specific expression of RUNX2, a key transcription factor involved in the rapid regeneration of new bone [77].

### 4.4. Dental Implants

In addition to existing or potential orthopedic uses, silicon nitride is an emerging material for dental restoration and dental implant applications [11,12,78]. This is grounded in its superior mechanical, thermal, tribological, biological, and antibacterial properties. Silicon nitride is considered an attractive ceramic for dental applications with good mechanical properties even in its porous form. In addition, it has advantages over conventional ceramics used as restorative materials, including its inherent antibacterial/anti-infective activity, radiolucency, and lower hardness [79].

Although zirconia today is still most commonly used for dental implants, onlays, and veneers, its intrinsic bioinert nature frequently counteracts biological integration. A silicon nitride-based laser-cladding process has been developed to improve the biological response to biomedical zirconia by forming a composite coating with silicon nitride particles dispersed in a nanocrystalline/amorphous silicon matrix. The bone tissue quality was found to be comparable to healthy human bone tissue, suggesting that laser cladding with silicon nitride might be able to improve the stability of zirconia dental implants [80].

However, at present, there are no clinical studies and trials yet to confirm the positive changes silicon nitride may elicit as a dental implant material. Further research and clinical trials are needed into the in vivo performance of antibacterial surface coatings, addressing issues such as enhancing bone regeneration around the implant surface or reducing the risk of periimplantitis [11]. Nevertheless, the ceramic nature of silicon nitride makes it a promising candidate for dental implantology since the release of cytotoxic metallic by-products can be avoided. Higher esthetic appeal, improved osseointegration and biocompatibility, as well as antibacterial activity together with high wear and corrosion resistance render silicon nitride a powerful contender for dental implant applications.

### 4.5. Antibacterial and Antiviral Applications

Silicon nitride ceramics show no cytotoxic effects, confirming their high degree of biocompatibility as ascertained by cell culture tests with mouse fibroblasts (L929) and human mesenchymal stem cells (hMSCs) [81]. 

Silicon nitride was found to efficiently combat Gram-negative bacteria, presumably based on chemically driven mechanisms that are related to the pH-dependent surface chemistry of silicon nitride. The antiviral property of Si_3_N_4_ derives from a hydrolysis reaction at its surface and the subsequent formation of reactive nitrogen species in doses that could be metabolized by mammalian cells but are lethal to pathogens [82]. Since the presence of these reactive nitrogen species creates osmotic stress in the cytoplasmic space, the metabolic rate changes rapidly, the bacterial membrane is damaged, and fast lysis occurs [83]. Silicon nitride surfaces were found to inactivate the SARS-CoV-2 virus by several mechanisms that induce post-translational oxidative modifications of S-containing amino acids, the nitration of the tyrosine residue in the spike receptor binding domain, and the oxidation of RNA purines to form formamidopyrimidine. These modifications create molecular damage that in turn leads to a deleterious reshuffling of the secondary protein structure [84].

### 4.6. Photonic Application in Medical Diagnostics

Lasers for medical applications operate in a wide range of the electromagnetic spectrum, from X-ray to mid-IR. These electromagnetic waves need to be transmitted from the laser source to the target tissue by a waveguide, which will enable the easy manipulation of the laser beam in a medical diagnostic setting [85]. Photonic properties of silicon nitride, such as its high transparency at wavelengths below 1.1 µm, the ultra-low two-photon absorption effect at telecommunication wavelengths, and small propagation losses, allow building such waveguides that can manipulate and direct laser beams used in medical diagnostics. Based on silicon nitride waveguides, integrated nano-photonic sensing devices operating in the visible and near-infrared regions with drastically reduced propagation losses were developed [86].

Mammalian cell lines such as 3T3-L1 adipocytes and H9c2 cardiac myocytes were found to adhere strongly to silicon nitride surfaces. This property allows the visual investigation of their proliferation and phenotypic growth by fixing the cells on silicon nitride membranes for imaging and mapping across the electromagnetic spectrum, from X-ray fluorescence to X-ray absorption near-edge spectroscopy to visible and infrared micro-spectroscopies [87].

## 5. Conclusions

During the past 60 years, much research has been performed on silicon nitride as a multifunctional ceramic material. Initially, it was applied for its extraordinary mechanical, thermal, and chemical properties in a variety of industrial segments, reaching from automotive, aerospace, and metal forming and casting industries to semiconducting chip processing and parts for electronic systems. Only comparatively recently, the outstanding osseoinductive and antimicrobial properties of silicon nitride were discovered and utilized in several medical disciplines, and many applications were successfully developed [8]. In this contribution, several contemporary medical applications of silicon nitride were highlighted, including intervertebral spacers, orthopedic and dental implants, antibacterial and antiviral applications, and devices for medical diagnostics. 

However, much additional work is still required to improve and enhance the inherent biocompatibility of the material. A fertile field of research exists to make silicon nitride parts and coatings fit for bearing surfaces of joint implants. Such bearings will benefit from the very low wear rate, long-term mechanical resilience, and low coefficient of friction of silicon nitride ceramics. However, since the high modulus of elasticity of bulk silicon nitride is still counter indicative for its use for structural parts of arthroplastic implants, it cannot compete successfully with classic austenitic stainless steels and CoCrMo or titanium alloys at this time. Here, strong and well-adhering silicon nitride-based coatings may come to the rescue that would, on the one hand, allow the use of low-modulus substrate materials to satisfy the biomechanical requirement to avoid stress shielding and, on the other hand, provide osseoinductivity and antibacterial protection to a variety of future implants. Thus, future medical implants coated with silicon nitride will combine important biomechanical and biological properties in a synergistic way. 

## Figures and Tables

**Figure 1 materials-16-05142-f001:**
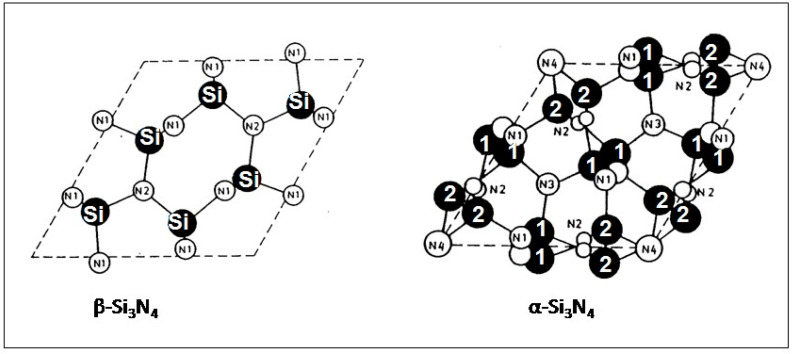
Structures of hexagonal β- and trigonal α-silicon nitride. Projection down the c-axis onto the AB plane. Black circles: silicon atoms, white circles: nitrogen atoms [18]. © With permission from Wily-VCH, Weinheim, Germany.

**Figure 3 materials-16-05142-f003:**
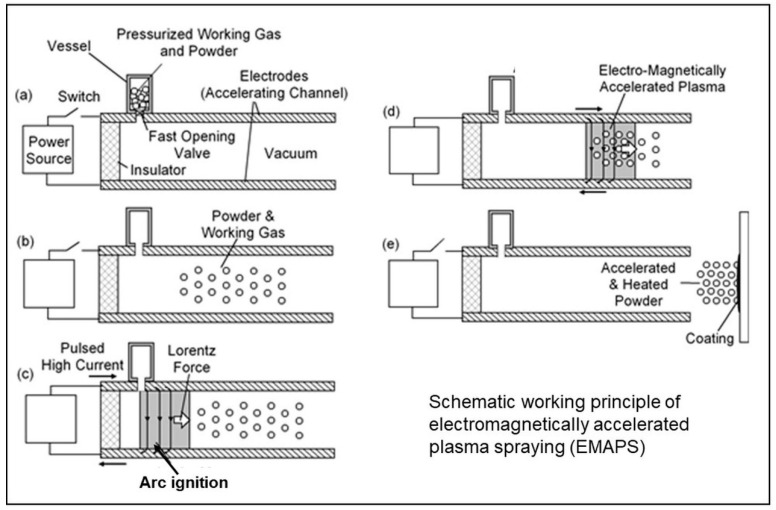
Schematic working principle of EMAPS [51]. © With permission from ASM International. (**a**) Equipment configuration (**b**) Introduction of powder particles and working gas into the acceleration channel (**c**) Arc discharge creates electromagnetic force pulse (**d**) Formation of electromagnetically accelerated plasma (**e**) Deposition of coating.

**Figure 4 materials-16-05142-f004:**
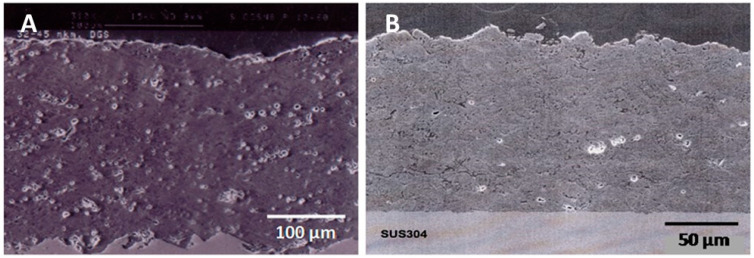
Silicon nitride-based coatings on steel produced by detonation spraying (**A**) [46] and EMAPS (**B**) [51]. © With permission from ASM International.

**Figure 5 materials-16-05142-f005:**
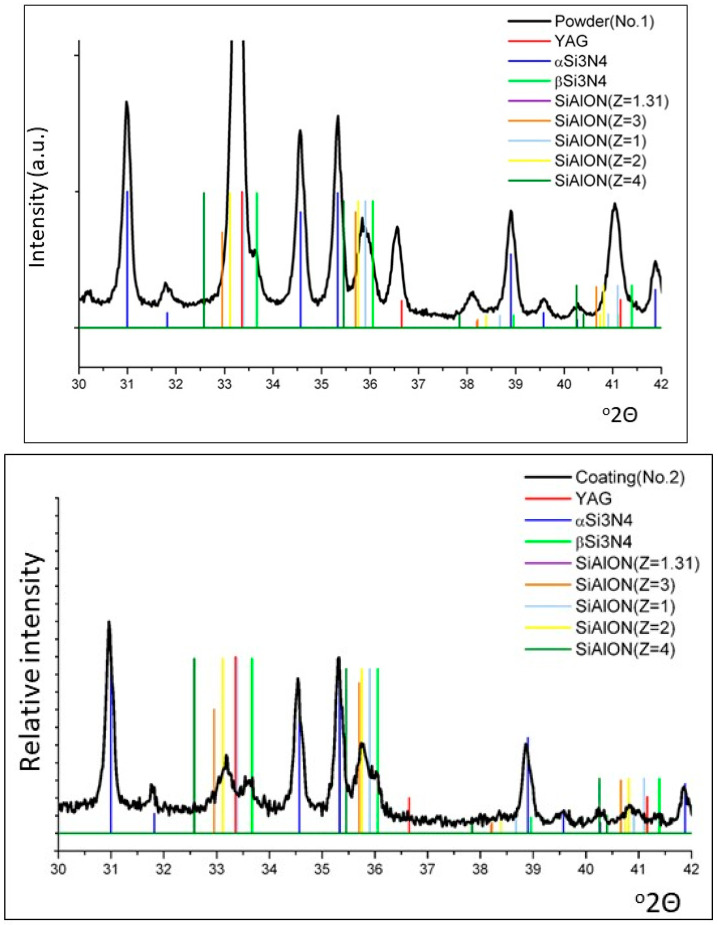
Phase composition of precursor powder #1 (**top**) and the resulting EMPA-sprayed silicon nitride coating (**bottom**). © Image courtesy of Dr. Shu Usuba, National Institut of Advanced Industrial Science and Technology, Tsukuba, Japan.

**Table 1 materials-16-05142-t001:** Desired characteristics of silicon nitride nanoscale particles [18].

Nanoscale Particle Properties	Desired Characteristics
Average particle size	1–250 nm
Grain size distribution	Narrow or broad, depending on application
Particle shape	Spheres, needles, platelets, and wires
Degree of agglomeration	None or controlled
Purity level	Very high
Dispersability	High, in water and solvents ^1^
Production scale	Several tons/year
Reproducibility	Very high (6σ)
Safety	Inherent, contained
Economy	5 to 250 EUR/kg
Patents	Well established

^1^ Solvents may include ethanol, acetone, and organic surfactants such as silicone compounds.

**Table 2 materials-16-05142-t002:** Selected properties of RBSN, SSN, SRBSN, and TSSN [18].

Property	RBSN	SSN	SRBSN	TSSN
Density (Mg∙m^−3^) (% of theoretical density)	70–88	95–100	~99	96–100
Flexural strength (4-point, 25 °C) (MPa)	150–350	500–1000	850	1400
Fracture toughness (25 °C) (MPa∙√m)	1.5–3	5–8	11	>14
Fracture energy (J∙m^−2^)	4–10	~60		
Modulus of elasticity (25 °C) (GPa)	120–220	300–330		280–540
Thermal conductivity (25 °C) (W/mK)	4–30	15–50	>170	
Thermal shock resistance R (K)	220–580	300–780		
Thermal shock fracture toughness R’ (W∙m^−1^)	500–10,000	7000–32,000		
Coefficient of thermal expansion (10^−6^ K^−1^)	3.2	3.2		>3

## Data Availability

Data are available on request.

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
