# Peer review of "Silicon Nitride Ceramics: Structure, Synthesis, Properties, and Biomedical Applications"

_materials, 2023, doi:10.3390/ma16145142_

Round 1
Reviewer 1 Report
For the purpose of various application, the introduction should highlight the exceptional qualities of the silicon nitride ceramics
In Table 1, the type of solvent used in the “dispersability” should be specified.
Figure 2 should illustrate the numerous relationships between processing routes.
Line 170-178: The statement you provided seems to be accurate based on the information given. It states that the development of sintered reaction-bonded silicon nitride (SRBSN) has been undertaken to reduce the porosity of RBSN (reaction-bonded silicon nitride) and enhance its mechanical and elastic properties. Recent advancements in SRBSN have demonstrated improvements in toughness and impact resistance by promoting the growth of rod-like silicon nitride grains that can arrest or deflect cracks. The use of oxidic sintering aids and specific processing conditions can also lead to high thermal conductivities. The text further mentions that the addition of AlN to Y2O3 as a sintering aid, along with impurity oxygen, can dissolve in the lattice of β-Si3N4 and form a solid solution called β-SiAlON (Si6-zAlzOzN8-z). Additionally, it suggests that increasing the sintering time can enhance fracture toughness. The references [22] and [23] are provided to support these claims. Table 2 should be added in this paragraph.
Many paragraphs lack citations of sources.
Almost all illustrations and data If they belong to the author, they should be revised to improve readability and eliminate redundancy.
Eq 3: The equation you provided represents a chemical reaction involving silicon nitride (Si3N4), aluminum oxide (Al2O3), and yttrium oxide (Y2O3). The reaction is showing the formation of a phase called β’-Si6-zAlzOzN8-z, which is a solid solution of silicon, aluminum, oxygen, and nitrogen. The other product mentioned is Y-Si-Al-O-N glass, which refers to a glassy phase containing yttrium, silicon, aluminum, oxygen, and nitrogen.
While the equation itself is not explicitly stated as true or false, it does represent a possible reaction that can occur under certain conditions. It suggests the formation of specific phases when Si3N4, Al2O3, and Y2O3 react together. The actual reaction and the formation of the mentioned phases would depend on factors such as temperature, pressure, and the specific composition and ratios of the starting materials. It's important to note that the equation you provided is a simplified representation and may not capture the complete complexity of the reaction system.
Line 311-317: To address this issue, future development cycles should focus on process scale-up and optimization to enable a continuous injection mode of the pressurized gas and powder particles. By achieving a continuous injection mode, the process can become more efficient and cost-effective, improving its viability from a process economy perspective.
Updating the information to include these points would provide a more comprehensive and accurate understanding of the challenges and potential areas for improvement in EMAPS coatings.
The article lacks sufficient information, multiple examples, and comprehensive reference materials.
Some of the most important technical terms should be defined in the article.
Reviewer 2 Report
The authors reviewed the structure, properties, and application of silicon nitride (SN) and discuss their relationship. First, the crystallographic structures of the SN were presented, followed by the synthesis methods for different SN forms, including powders, compacts, foams, and coatings. Finally, different biomedical applications of the SN products such as intervertebral spacers, orthopedic applications, craniofacial reconstruction, dental implants, antibacterial and antiviral applications, and photonic application in medical diagnostics were discussed. The work is interesting and can be published in Materials if the following issues can be addressed:
1. Abbreviations should be defined once when they are first mentioned in the manuscript. VLSI, R.F., PLA, and PMMA were not defined. CVD, SSN, RBSN, SRBSN, TSSN, TGG, CPA, EA, TCA, SMFA, RMFA, EMAPS, and ECM were defined several times. OLED, MSRA, PECVD, SHS, DS, APS, HFPD, ECF, ALIF, FEA, HR, COF, UHMWPE, hMSCs, RNS, XANES, and XRF were mentioned only once and should not be defined. The authors should correct the issues in the revised manuscript.
2. In table 1, why were narrow and broad grain size distributions desired?
3. More references are required for the routes toward the synthesis of SN powders.
4. In table 2, why was the flexural strength of the TSSN the best?
5. More references are required for the SN foam sections.
6. In Figure 4, why were only the morphologies of the SN coatings for EMAPS presented in this work? How about the morphologies of the other methods?
7. Was Figure 4 obtained from the author’s work? If it was from the work of other studies, the author should add references in its caption and obtain permission to reuse it.
8. The author mentioned in abstract that potential future uses are outlined. However, this part was not clearly shown in the manuscript.
9. Conclusion should be improved with a summary of this work.
Some grammar errors in the manuscript need to be corrected.
Round 2
Reviewer 1 Report
No comments
No comments
Author Response
No comments to respond to.,